# Efficacy and Safety of First-Line Cytokines versus Sunitinib and Second-Line Axitinib for Patients with Metastatic Renal Cell Carcinoma (ESCAPE Study): A Phase III, Randomized, Sequential Open-Label Study

**DOI:** 10.3390/cancers15102745

**Published:** 2023-05-13

**Authors:** Yoshifumi Kadono, Hiroyuki Konaka, Takahiro Nohara, Kouji Izumi, Satoshi Anai, Kiyohide Fujimoto, Tomoyuki Koguchi, Kei Ishibashi, Noriyasu Kawai, Keita Nakane, Akinori Iba, Naoya Masumori, Shizuko Takahara, Atsushi Mizokami

**Affiliations:** 1Department of Integrative Cancer Therapy and Urology, Kanazawa University Graduate School of Medical Science, Kanazawa 920-8640, Japan; 2Department of Urology, Japanese Red Cross Society Kanazawa Hospital, Kanazawa 921-8162, Japan; 3Department of Urology, Nara Medical University, Kashihara 634-8522, Japan; 4Department of Urology, Fukushima Medical University School of Medicine, Fukushima 960-1295, Japan; 5Ishibashi Urology Clinic, Koriyama 963-8002, Japan; 6Department of Nephro-urology, Nagoya City University Graduate School of Medical Sciences, Nagoya 467-8601, Japan; 7Department of Urology, Gifu University Graduate School of Medicine, Gifu 501-1194, Japan; 8Department of Urology, Wakayama Medical University, Wakayama 641-8509, Japan; 9Department of Urology, Sapporo Medical University School of Medicine, Sapporo 060-8556, Japan; 10Innovative Clinical Research Center (iCREK), Kanazawa University Hospital, Kanazawa 920-8641, Japan

**Keywords:** cytokine, interferon alfa, interleukin-2, sunitinib, axitinib, metastatic renal cell carcinoma

## Abstract

**Simple Summary:**

This was a phase III randomized controlled trial investigating the outcomes of low-dose interleukin-2 (IL-2) plus interferon alfa (IFNα) versus sunitinib as the first line and axitinib as the second line in patients with low- and intermediate-risk metastatic renal cell carcinoma (mRCC). There was a trend toward better total progression-free survival up to the end of the second line for IL-2 + IFNα but no significant advantage in terms of overall survival. The study was underpowered to draw any definitive conclusions. The results showed no clear advantage of IL-2 + IFNα over sunitinib in the first line; however, it may be an option in some relatively low-risk mRCC cases due to the difference in the adverse events profile. This trial was registered with the University Hospital Medical Information Network (UMIN), center identifier UMIN 000012522.

**Abstract:**

Background: The sequence of first-line cytokine and second-line molecular targeted therapies may be suitable for some patients with metastatic renal cell carcinoma (mRCC) because of the expectation of complete remission and durable response achieved with cytokine therapy. Methods: This was a phase III randomized controlled trial investigating the outcomes of low-dose interleukin-2 (IL-2) plus interferon alfa (IFNα) versus sunitinib as the first line and axitinib as the second line in patients with low- and intermediate-risk mRCC. Results: Thirty-five patients were randomly assigned. The total progression-free survival (PFS) to the end of the second line was 29.0 months (95% CI, 11.7–46.3) in the IL-2 + IFNα group and 16.3 months (95% CI, 6.3–26.4) in the sunitinib group. The PFS hazard ratio for the IL-2 + IFNα group relative to the sunitinib group was 0.401 (95% CI, 0.121–1.328; *p* = 0.135). The hazard ratio for overall survival (OS) was 1.675 (95% CI, 0.418–6.705; *p* = 0.466), which was better in the sunitinib group than in the IL-2 + IFNα group but not statistically significant. The types of adverse events (AEs) differed significantly, although there was no significant difference in the incidence of AEs. Conclusions: There was a trend toward better total PFS for IL-2 + IFNα, but it was not significant. There was also no advantage of IL-2 + IFNα in terms of OS. The study was underpowered to draw any definitive conclusions. The results showed no clear advantage of IL-2 + IFNα over sunitinib in the first-line setting; however, it may be an option in some relatively low-risk mRCC cases due to the difference in the AE profile. This trial was registered with the University Hospital Medical Information Network (UMIN), center identifier UMIN 000012522.

## 1. Introduction

Treatment strategies for metastatic renal cell carcinoma (mRCC) have changed from cytokines to molecularly targeted drugs and immune checkpoint inhibitors because of their superior efficacy. These new therapies have improved the probability of complete recovery from mRCC [1,2,3], but the rates are still low [4,5,6,7]. Therefore, the realistic goal of current mRCC treatment is to maximize overall survival (OS) while maintaining the quality of life of patients. Achievement of this goal requires a long-term treatment strategy that takes into account the general condition of the patient and control of treatment-related adverse events (AEs). However, appropriate protocols for sequential treatment of mRCC have not yet been established. The number of treatment lines has increased to secondary, tertiary, and quaternary, and the number of patients who can receive treatment is gradually decreasing due to disease progression and the occurrence of drug-induced AEs that worsen the overall condition of the patients. The optimal choice of a first-line drug and a contiguous second-line drug is very important for prolonging the OS of patients with mRCC. Judging from the guidelines for mRCC treatment and the Japanese system of national insurance, the recommended treatment regimen for mRCC categorized as having favorable risk by the Memorial Sloan Kettering Cancer Center (MSKCC) criteria [8,9] is sunitinib [1] or pazopanib [3] as a first-line drug and axitinib [2] or nivolumab [10] as a second-line drug. The current guidelines still recommend these treatments in the first-line setting, especially for patients with a favorable risk profile [11]. However, the OS of patients treated with cytokines, which have long been used for mRCC in Japan, has been reported to be better than that in western countries [12]. Therefore, it is presumed that the efficacy of cytokine therapy is better in Japanese patients than in patients in western countries. Among mRCC cases meeting the favorable risk criteria, complete remission (CR) and sustained response have been reported using low-dose interleukin-2 (IL-2) plus interferon alfa (IFNα) [13,14]; However, molecular targeted therapy may not achieve CR in mRCC patients. Cytokine therapy is not effective for high-risk mRCC patients but may be effective for some intermediate-risk mRCC patients because they are part of a heterogeneous group [15].

We conducted a prospective randomized controlled trial (RCT) for patients with favorable- and intermediate-risk mRCC (classified by MSKCC risk criteria) to evaluate the efficacy and safety of sequential treatment of cytokine (IL-2 + IFNα) as the first-line and axitinib as the second-line therapy versus sequential treatment of sunitinib as the first-line and axitinib as the second-line therapy, which is one of the standard treatments for favorable-risk mRCC (ESCAPE study) [16].

## 2. Materials and Methods

### 2.1. Patients

The study population consisted of patients who were aged 20–80 years and had clear cell mRCC. The inclusion criteria were: patients with no previous history of systemic therapy for mRCC; MSKCC risk criteria of favorable or intermediate; Eastern Cooperative Oncology Group (ECOG) performance status of 0 or 1; and adequate hematologic, hepatic, and renal function [16]. Written informed consent was obtained from all patients.

### 2.2. Study Design

This was a phase III, investigator-initiated, multicenter RCT involving a head-to-head comparison of IL-2 + IFNα vs. sunitinib as 1st-line therapy followed by axitinib as 2nd-line therapy for patients with mRCC. Eligible patients were randomly assigned to the 1st-line IL-2 + IFNα or sunitinib treatment groups by Waritsukekun (Mebix, Tokyo, Japan) using a stratified sampling method to obtain adequate between-group balance with respect to age (< 70 or > 70 years), MSKCC risk profile (favorable or intermediate), status of metastasis (lung only or other than lung), and participating institution. Patients assigned to the cytokine group received an intravenous injection of IL-2 at a dose of 700,000 unit/day for five days/week for the initial 2 or 4 weeks and then one or three times/week as maintenance plus subcutaneous injection of 5–6 million units of IFNα two or three times/week. Patients in the sunitinib group received oral sunitinib at a dose of 50 mg/day, and a single treatment course consisted of 4 weeks of administration and 2 weeks of withdrawal. A schedule of 2 weeks of administration and one week of withdrawal was also permitted. Reduction of doses was permitted if the clinical investigator considered the basic doses inappropriate for any reason. Administration of IL-2 + IFNα or sunitinib was terminated upon confirmation of progressive disease (PD) based on Response Evaluation Criteria in Solid Tumors (RECIST) v.1.1, death, or occurrence of severe AEs. A clinical investigator administered axitinib as 2nd-line treatment immediately after confirmation of PD based on RECIST v.1.1 during 1st-line treatment. In case of cessation of 1st-line treatment because of severe AEs, a clinical investigator was allowed to consider restarting 1st-line treatment after recovery below grade 2. Axitinib at a dose of 10 mg/day (5 mg dose twice per day) was orally administered to patients as 2nd-line treatment. Dose reduction or dose increase up to 20 mg/day (10 mg dose twice per day) was permitted if a clinical investigator considered the basic doses as inappropriate for any reason. Administration of axitinib was terminated upon confirmation of PD based on RECIST v.1.1, death, or occurrence of severe AEs corresponding to discontinuation criteria [16]. This study was designed and registered at the Trial University Hospital Medical Information Network (UMIN), center identifier UMIN 000012522, in July 2013.

### 2.3. Efficacy and Safety Assessment

The primary endpoint was total progression-free survival (PFS) from randomization to PD based on RECIST v.1.1 criteria examined with CT or MRI or death during the therapy. PD during 1st-line therapy was not counted as an event based on the definition. In case of the cancellation of 1st-line therapy due to severe AEs or refusal of the patient without confirmation of PD, the end date of 1st-line therapy was not counted as cessation, and the case continued to receive the 2nd-line therapy. Secondary endpoints included OS (defined as the time from randomization to death from any cause), each PFS in 1st-line treatment and 2nd-line treatment (based on RECIST v.1.1 criteria), objective response rate (ORR) (defined as the proportion of CR and partial response (PR) cases in total cases based on best overall response using RECIST v.1.1 criteria), disease control rate (DCR) (defined as the proportion of CR, PR, and stable disease (SD) cases in total cases in 1st-line and 2nd-line treatment), and safety in 1st-line and 2nd-line treatment evaluated by Common Terminology Criteria for Adverse Events v.4.0 system. All patients were asked to complete a medical history. Clinical data obtained in the ESCAPE study included the ECOG performance status, physical examination findings, results of hematologic examination, blood biochemical examination, urinalysis, chest X-ray imaging, lung-to-pelvic CT or MRI, brain CT or MRI, bone scintigraphy, and electrocardiography. X-ray, brain CT, and bone scintigraphy were performed at the time of study registration. Other examinations were performed every 2 months from the date of the commencement of treatment up to month 24 and every 3 months after month 24 until the completion of the study [16].

### 2.4. Statistical Analysis

In the AXIS trial (2nd-line axitinib versus sorafenib) reported by Rini et al. [2] and the reports by Akaza et al. [17], the median total PFS was 26.1 (10.4 + 15.7) months for a combination of IL-2 + INFα and axitinib and 15.3 months (10.5 + 4.8) for the combination of sunitinib and axitinib. Based on these reports, in this study, the hypothetical median PFS was set as 26 months for the combination of IL-2 + INFα and axitinib and 15 months for the combination of sunitinib and axitinib. The sample size was calculated factoring a study duration of 5 years (two-year entry period and three-year observation period) and the difference in total between the groups. To detect a significant difference between the groups by log-rank test with significance level of 0.05 and power of 80%, at least 65 patients were required in each group. Furthermore, factoring a dropout rate of approximately 10% for various reasons, the target sample size was set at 72 patients per group (total 144 cases) [16].

Treatment-based analyses in each group were performed, and survival curves were estimated using the Kaplan–Meier method. Between-group differences with respect to survival outcomes were assessed using log-rank test. Hazard ratios were estimated using the Cox proportional hazard model. ORR (CR + PR) and DCR (CR + PR + SD) in each group were evaluated using the best response evaluated by RECIST v1.1. AEs in each group were aggregated by grades and types, and the between-group differences in the proportion of AEs were assessed using the Fisher’s exact test. Two-sided *p*-values < 0.05 were considered indicative of statistical significance for all tests.

## 3. Results

### 3.1. Patient Characteristics

Between October 2013 and September 2017, 35 patients were randomly assigned (18 patients in the IL-2 + IFNα group and 17 patients in the sunitinib group) at 10 hospitals in Japan (Figure 1). Two patients in the sunitinib group did not receive sunitinib, while 18 patients in the IL-2 + IFNα group and 15 patients in the sunitinib group received at least one dose of the drug and were analyzed for safety (Figure 1). The treatment groups were balanced with respect to baseline demographic and disease-related characteristics (Table 1). Excluding the ineligible cases, 17 patients in the IL-2 + IFNα group and 13 patients from the sunitinib group were selected for efficacy analysis (Figure 1). The study period ended in September 2020, and the results were analyzed.

### 3.2. Treatment Duration

The median duration of first-line treatment was 7.8 months (range 0.9–57.5) in the IL-2 + IFNα group and 9.0 months (range 0.4–47.0) in the sunitinib group. First-line treatment with IL-2 + IFNα was continued in two patients (12%) until the last observation period; however, the sunitinib group did not have such cases (Figure 1). During the course of treatment, consent was withdrawn by one patient in the IL-2 + IFNα group and three patients in the sunitinib group. In the IL-2 + IFNα group, 13 of the 15 patients who discontinued first-line treatment received axitinib, and, in the sunitinib group, 5 of the 13 patients who discontinued first-line treatment received second-line axitinib. In the IL-2 + IFNα group, 17 patients were included in the analysis of treatment efficacy: two patients who continued on first-line treatment, thirteen patients who progressed to second-line treatment with axitinib, and two patients who chose a treatment other than axitinib. In the sunitinib group, 13 patients were included in the analysis of treatment efficacy: one patient who died of other causes during first-line treatment, five patients who progressed to second-line treatment with axitinib, and seven patients who were selected for treatment other than axitinib. For safety evaluation, 18 patients in the IL-2 + IFNα group and 15 patients in the sunitinib group who received at least one dose were included in the analysis (Figure 1). The median duration of second-line axitinib was 15.4 months (range 1.5–49.0) in the IL-2 + IFNα group and 2.9 months (range 1.7–25.2) in the sunitinib group. Axitinib in the second line was continued in two patients in the IL-2 + IFNα group compared to none in the sunitinib group. The 14 patients who progressed on axitinib in the second line (nine in the IL-2 + IFNα group and five in the sunitinib group) proceeded to the third line and beyond. One of the two patients in the IL-2 + IFNα group who did not receive second-line treatment with axitinib received pazopanib as a second-line treatment, and the other did not receive second-line treatment. Three of the seven patients in the sunitinib group who did not receive second-line treatment with axitinib received second-line treatment with another agent (one each with everolimus, nivorumab, and pazopanib). The remaining four patients did not receive second-line therapy (Figure 1).

### 3.3. Adverse Events

All patients in both groups experienced treatment-related AEs during first-line treatment (Table 2). The incidence of ≥G3 AEs in the IL-2 + IFNα group was significantly lower than that in the sunitinib group (8 patients (44%) and 13 patients (87%), respectively; *p* = 0.012); however, G4 AEs were observed only in the IL-2 + IFNα group (four patients (27%)). AEs that tended to be more common in the IL-2 + IFNα group in all grades were pyrexia (72% vs. 33%, *p* = 0.025) and malaise (50% vs. 27%, *p* = 0.172). The common AEs in the sunitinib group were hypertension (47% vs. 11%, *p* = 0.022), diarrhea (27% vs. no cases, *p* = 0.019), hypothyroidism (33% vs. no cases, *p* = 0.008), palmar–plantar erythrodysaesthesia (53% vs. no cases, *p* < 0.001), stomatitis (27% vs. no cases, *p* = 0.019), and decreased platelet count (60% vs. 17%, *p* = 0.010). The incidence of decreased appetite (IL-2 + IFNα vs. sunitinib: 33% vs. 20%, *p* = 0.392), anemia (22% vs. 13%, *p* = 0.510), proteinuria (11% vs. 13%, *p* = 0.846), decreased neutrophil count (39% vs. 40%, *p* = 0.948), and decreased white blood cell count (44% vs. 40%, *p* = 0.797) was similar in both groups. Regarding ≥ G3 AEs, those more frequently observed in the sunitinib group were hypertension (40% vs. 5%, *p* = 0.016), proteinuria (13% vs. no cases, *p* = 0.110), and decreased platelet count (40% vs. no cases, *p* = 0.003). The incidence of a ≥ G3 decrease in neutrophil count (17% vs. 7%, *p* = 0.381) and white blood cell count (17% vs. 7%, *p* = 0.381) tended to be higher in the IL-2 + IFNα group, but the between-group difference was not statistically significant. Second-line axitinib AEs were generally similar between the two groups (Table 2).

### 3.4. Objective Response Rate, Disease Control Rate, and Progression-Free Survival with each First-Line and Second-Line Treatment

There was no significant difference in first-line ORR (IL-2 + IFNα 43% vs. sunitinib 31%, *p* = 0.474), but the DCR tended to be better with sunitinib (IL-2 + IFNα 81% vs. sunitinib 100%, *p* = 0.099), with no cases of PD (Table 3). Both the ORR and DCR with second-line axitinib were better in the IL-2 + IFNα group, but statistically significant differences were seen only in ORR (ORR; IL-2 + IFNα 62% vs. sunitinib 0%, *p* = 0.019, DCR; IL-2 + IFNα 85% vs. sunitinib 60%, *p* = 0.261) (Table 3). First-line PFS tended to be worse in the IL-2 + IFNα group (median: 9.9 months (95% CI, 5.1–14.7)) than in the sunitinib group (median: 14.5 months (95% CI, 6.3–22.7)), but the between-group difference was not statistically significant (Figure 2A). The PFS hazard ratio for the IL-2 + IFNα group relative to the sunitinib group was 1.409 (95% CI, 0.560–3.545; *p* = 0.466). PFS for second-line axitinib tended to be better in the IL-2 + IFNα group (median: 14.7 months (95% CI, 2.5–26.9)) than in the sunitinib group (median: 3.7 months (95% CI, 0.0–8.2)), but the between-group difference was not statistically significant. For second-line axitinib, the PFS hazard ratio for the IL-2 + IFNα group relative to the sunitinib group was 0.595 (95% CI, 0.198–1.787; *p* = 0.355) (Figure 2B).

### 3.5. Total (First Line + Second Line) Progression-Free Survival

The total PFS from the start of the first line to the end of the second line was analyzed in 13 patients in the IL-2 + IFNα group and 5 patients in the sunitinib group who received axitinib in the second line (29.0 months (95% CI, 11.7–46.3) in the IL-2 + IFNα group versus 16.3 months (95% CI, 6.3–26.4) in the sunitinib group). The hazard ratio for the IL-2 + IFNα group relative to the sunitinib group was 0.401 (95% CI, 0.121–1.328; *p* = 0.135) (Figure 3A). After the first line, the study included four patients in the IL-2 + IFNα group and eight patients in the sunitinib group who did not receive axitinib according to the protocol (Figure 1) for a total of 30 patients (17 in the IL-2 + IFNα group and 13 in the sunitinib group). The first-line (+ second-line) PFS for these patients was similar in the two groups (median: 30.3 months (95% CI, 22.4–38.2) in the IL-2 + IFNα group versus 27.6 months (95% CI, 13.9–41.4) in the sunitinib group; *p* = 0.749) (Figure 3B).

### 3.6. Overall Survival

During the observation period, the median OS was not reached in either of the groups. The mean OS in the IL-2 + IFNα group and sunitinib group was 59.5 months (95% CI, 45.7–73.4) and 63.8 months (95% CI, 53.1–74.6), respectively. The hazard ratio was 1.675 (95% CI, 0.418–6.705; *p* = 0.466), indicating better OS in the sunitinib group than in the IL-2 + IFNα group, but the difference was not statistically significant (Figure 4). The 1-year, 3-year, and 5-year survival rates were 94% vs. 92%, 75% vs. 92%, and 60% vs. 81% for the IL-2 + IFNα and sunitinib groups, respectively (Figure 4).

## 4. Discussion

In this study, as initially expected, the effect of axitinib in the second-line setting was better in the IL-2 + IFNα group than in the sunitinib group. In the first-line setting, the sunitinib group tended to do better than the IL-2 + IFNα group, but the difference was not statistically significant. The combined results of both lines tended to be better in the IL-2 + IFNα group (Figure 3A). Judging from the AXIS trial reported by Rini et al. [2] and the reports from Akaza et al. [17], the probability of median total PFS (first line + second line) was calculated as 26.1 (10.4 + 15.7) months for a combination of IL-2 + INFα and axitinib and 15.3 months (10.5 + 4.8) for a combination of sunitinib and axitinib [16]. Although the total PFS was in line with expectations, the number of patients in the sunitinib group who progressed to axitinib in the second line was small, and some patients did not receive axitinib in the second line because they had been treated for a longer period in the first line. Therefore, in cases where patients did not receive axitinib in the second line, the total PFS was evaluated by adding the time up to the progression in the first line. As a result, the first-line (+ second-line) result in the sunitinib group was prolonged to a median of 27.9 months, resulting in no significant difference in total PFS in each group (Figure 3B). Perhaps reflecting this result, OS tended to be slightly better in the sunitinib group, although the difference was not statistically significant. The best tumor response at the first line was better in the sunitinib group, with no PD cases and better DCR in the sunitinib group, although the ORR was not significantly different between the two groups (Table 3). Second-line axitinib response was better in the IL-2 + IFNα group, as expected, with significant differences in ORR (Table 3).

AEs of all grades occurred in all patients in both groups; ≥ G3 AEs were significantly more common in the sunitinib group, but G4 AEs were observed only in the IL2 + IFNα group. The profile of side effects was different in each group, and the characteristics of each were similar to those reported in previous reports [1,18]. Second-line axitinib AEs did not appear to be affected by first-line treatment.

According to the published guidelines for mRCC during the period of this clinical study, for favorable risk classified by MSKCC criteria [9] or the International mRCC Database Consortium criteria (IMDC) [8], sequential treatment of sunitinib as first-line therapy [1] and nivolumab as second-line therapy [10] is recommended. For intermediate- and high-risk cases, sequential treatment with ipilimumab + nivolumab as first-line therapy [19] and tyrosine kinase inhibitors (TKI), including vascular endothelial growth factor (VEGF)-targeted drugs [2], is recommended with high-level evidence. Recent guidelines recommend combination therapies, mainly immuno-oncology (IO) drugs, as preferred regimens regardless of IMDC risk, and sunitinib is in the category of the recommended regimens for first-line therapy [11,20,21]. Additionally, cabozanitinib is recommended as an alternative second-line therapy after TKI failure [20,21]. In particular, the efficacy of IO + IO [4,19] and IO + TKI combinations [6,7,22,23] has been demonstrated in intermediate- to high-risk cases [24,25]. However, a recent network meta-analysis in the IMDC favorable-risk group found that first-line sunitinib is better than pembrolizumab + axitinib in terms of OS [24], and sunitinib alone is preferable in a certain number of patients as a first-line treatment. In addition, although high-dose IL-2 is mentioned as a treatment option, it is indicated as a limited choice, and the expert opinion does not clearly articulate the recommended cases [11]. IFNα is almost never mentioned in global guidelines [11,20,21]. However, IL2 + IFNα showed a favorable outcome in Japanese patients [17], and the response of TKI from IL2 + IFNα, which was expected to be superior to sunitinib in terms of therapeutic sequencing, was favorable [18]. The current clinical study was designed as a sequential treatment of first-line IL-2 + IFNα and second-line axitinib with the aim of outperforming sunitinib first-line therapy, especially for favorable-risk mRCC. In the current study, the sequence of IL-2 + IFNα and axitinib tended to compare favorably in terms of the total PFS with sunitinib and axitinib, as initially estimated. However, in actual practice, there was no clear superiority of the IL-2 + IFNα group over sunitinib in terms of both treatment efficacy and AEs, and the incidence of AEs was similar.

In previous guidelines, sunitinib was recommended as first-line treatment for favorable-risk mRCC [26,27], and it remains a recommended treatment option [11]. Sunitinib is a multi-targeted oral TKI that selectively inhibits signaling from receptor tyrosine kinases (RTKs) which contribute to tumor cell growth, tumor angiogenesis, and metastasis. Sunitinib inhibits the activity of RTKs such as platelet-derived growth factor receptor (PDGFR)-α and -β, VEGFR-1, -2, and -3, mast/stem cell growth factor receptor (c-kit), and FMS-like tyrosine kinase 3 [28,29]. In 2007, Motzer et al. reported the efficacy of sunitinib versus conventional therapy as first-line treatment for mRCC; an RCT of IFNα versus sunitinib in 750 patients with mRCC showed better PFS in the sunitinib group (11 months) compared to that in the IFNα group (5 months) (hazard ratio 0.42, *p* < 0.001) and better ORR in the sunitinib group (31% versus 6%) (*p* < 0.001) [1].

Axitinib is a selective and potent inhibitor of VEGFR-1, VEGFR-2, and VEGFR-3 activity and axitinib monotherapy is recommended as a second-line treatment option for mRCC [30,31]. VEGFR-1, VEGFR-2, and VEGFR-3 are major regulators of angiogenesis and lymphangiogenesis, and, in addition to the efficacy of axitinib against mRCC, its selectivity for VEGFR is expected to reduce toxicity caused by other TKIs that target multiple RTKs [30]. The AXIS trial, an RCT evaluating the efficacy of sorafenib versus axitinib in the second-line setting (715 patients with mRCC), reported a better median PFS in the axitinib group (6.8 months) than in the sorafenib group (4.7 months) (hazard ratio 0.665, *p* < 0.001) [2].

The ORR (CR + PR) for IL-2 or IFNα monotherapy has been reported to range from 10% to 20% [32]. The U.S. Food and Drug Administration approved high doses of IL-2 (600,000–700,000 IU/kg × 5 days for a 2-week cycle), but the IL-2 approved in Japan is a lower dose of 70,000–210,000 IU/kg/day [33,34]. Koreth et al. reported in 2011 that 1 million IU/day induced IL-2 regulatory T cells (Treg) for 2 to 3 months and prevented the development of graft-versus-host disease (GVHD) [35]. The usual dose of IL-2, 1.05 million IU/day, may contribute to enhancing the activity of natural killer cells (NK) and lymphokine-activated killer cells (LAK). However, it may induce Tregs and reduce antitumor effects such as NK and LAK activity. Since IFNα has been reported to suppress Tregs [36], combination therapy with low-dose IL-2 and IFNα is expected to contribute to antitumor activity without a reduction in Tregs. In the study by Akaza et al., 42 patients with mRCC were treated with low-dose IL-2 and IFNα. The reported ORR was approximately 35% (including two CR cases), DCR was approximately 70%, PFS was 10.4 months, and the 3-year OS rate was approximately 70%. These results compare favorably with IL-2 or IFNα monotherapy [17]. The main AEs of combination therapy are fever and general malaise, as shown in this study. Ito et al. reported that Japanese patients with mRCC who receive IFNα therapy live longer than their counterparts in western countries. Using genetic analysis of Japanese patients, they found a favorable effect of IFNα on mRCC allowing for long-term survival [37]. These reports suggest that cytokine therapy may contribute to the long-term survival of some mRCC patients, especially Japanese patients. Indeed, Naito et al. reported superior clinical outcomes for Japanese mRCC patients in the cytokine therapy era compared to outcomes reported in western countries [12,38]. Chow et al. reported that IL-2 treatment for mRCC showed an ORR of 48.1% and CR of 21.6% in selected patients with favorable pathology based on the composition of histologic growth patterns [39]. Cytokine therapy has the potential to achieve complete and durable responses in a limited number of mRCC patients. Thus, cytokine therapy may be the first-line treatment option of choice, especially for Japanese mRCC patients with favorable risk criteria.

In the AXIS study, the ORRs for sunitinib as first-line therapy versus cytokines as first-line therapy followed by axitinib as second-line therapy were 11.3% and 32.5%, respectively; the PFS was 4.8 months and 12 months, respectively. Thus, axitinib is more effective after cytokine therapy than after sunitinib [2]. In mRCC, the duration of response with first-line cytokine therapy and second-line TKI therapy may be longer than that with first-line TKI therapy and second-line therapy with other agents, especially in Japanese patients who are more likely to benefit from cytokine therapy. Conversely, sunitinib-induced AEs in Japanese patients tend to be more severe than that in western patients, and even those of us with accumulated experience of sunitinib AEs may have difficulty managing them. Sunitinib was initially associated with a high incidence of side effects, but it was later reported that both efficacy and AEs were favorable when the dosing regimen was changed from 4-week treatment with a 2-week rest period to 2-week treatment with a 1-week rest period [40]. 

There have been reports of long-term response to IL2 + IFNα [17], and this treatment sequence may be desirable for some patient populations; however, the number of patients enrolled in the present study was too small to identify the patient population that may benefit more with IL2 + IFNα therapy. The current study was underpowered to draw any definitive conclusions in this respect. After the initiation of this study in 2013, pazopanib, which is considered to cause fewer AEs than sunitinib, became available for first-line treatment in Japan in 2014 [41], and nivolumab became available for second-line treatment in 2016 [10]. The increasing number of patients opting for nivolumab after TKI administration may have also had an impact [42]. We speculate that the availability of these new drugs is a major reason for the lack of increase in the number of cases enrolled in this clinical study. In addition, IL-2 and IFNα are injectable preparations, and IL-2, in particular, is a preparation that cannot be self-injected, requiring frequent hospital visits for treatment. This may have led to poor patient acceptance of this clinical study in situations where treatment with oral agents, including sunitinib, was available.

## 5. Conclusions

In patients who were able to complete first-line IL-2 + IFNα and second-line axitinib and first-line sunitinib + second-line axitinib in this study, there was a trend toward better total PFS with IL-2 + IFNα and second-line axitinib; however, the difference was not statistically significant. Moreover, we observed no advantage of IL-2 + IFNα in terms of OS. The types of AEs in the two groups differed significantly, although there was no significant difference in the incidence of AEs. Owing to the small number of cases enrolled, this study was underpowered to draw any definitive conclusions. The results showed no clear advantage of IL-2 + IFNα over sunitinib in the first-line setting. However, IL-2 + IFNα in the first-line setting may be an option in some relatively low-risk mRCC cases due to the difference in AE profile. In addition, especially in the Japanese population, it may be optional as its effectiveness is better in the Japanese population than in the western population.

## Figures and Tables

**Figure 1 cancers-15-02745-f001:**
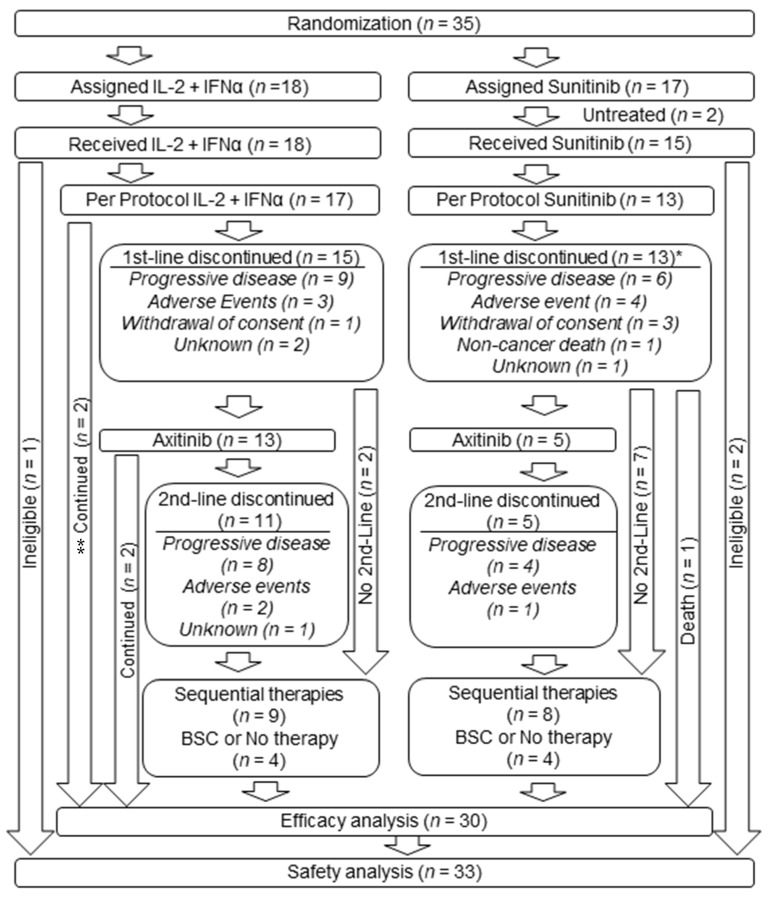
Patient enrolment and outcomes. * One case overlapped adverse events and withdrawal of consent, and the other overlapped adverse events and progressive disease. ** There were 4 cases without 2nd-line axitinib in the IL-2 + INFα group (2 cases of continuing 1st-line IL-2 + INFα and 2 cases of further treatment without 2nd-line axitinib) and 8 cases without 2nd-line axitinib in the sunitinib group (8 cases of further treatment without 2nd-line axitinib and 1 death case during 1st-line sunitinib treatment). BSC, best supportive care; IL-2, interleukin-2; INFα, interferon alfa.

**Figure 2 cancers-15-02745-f002:**
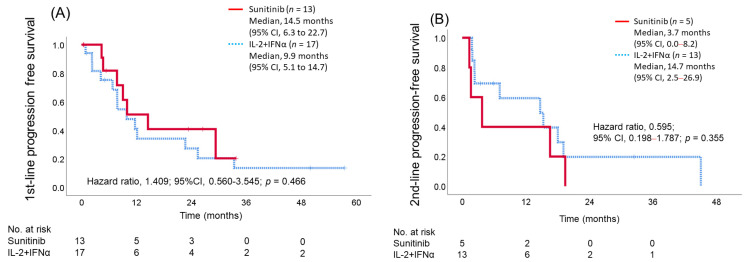
Kaplan–Meier estimate of 1st-line progression-free survival (**A**) and 2nd-line progression-free survival (**B**). IL-2, interleukin-2; INFα, interferon alfa.

**Figure 3 cancers-15-02745-f003:**
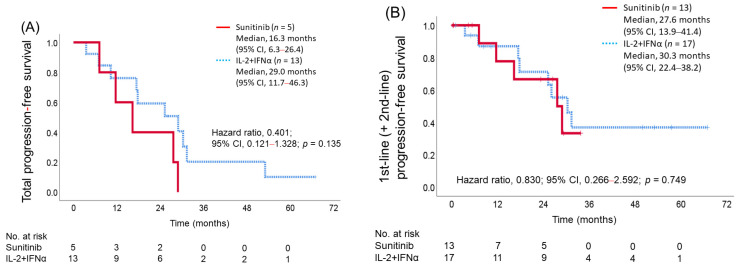
Total progression-free survival in 2nd-line treatment completed cases (**A**) and total progression-free survival (1st-line + 2nd-line treatment) including cases who did not receive 2nd-line treatment (**B**). IL-2, interleukin-2; INFα, interferon alfa.

**Figure 4 cancers-15-02745-f004:**
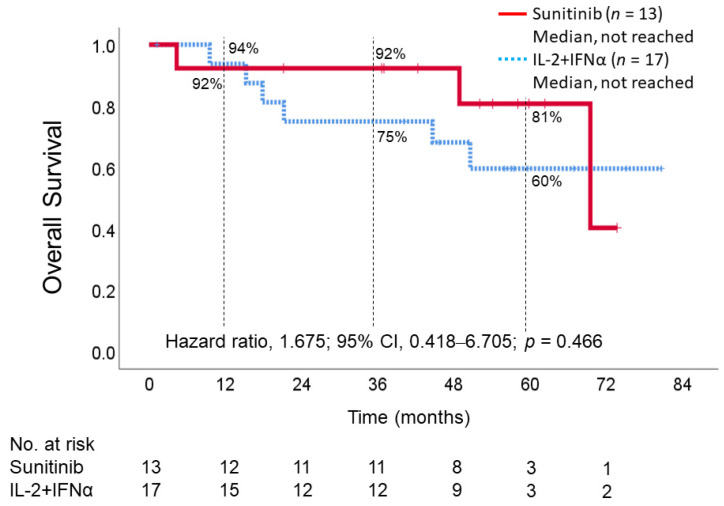
Kaplan–Meier estimate of overall survival. IL-2, interleukin-2; INFα, interferon alfa.

**Table 1 cancers-15-02745-t001:** Baseline demographic and clinical characteristics.

Variable	Median (Range), Number (%)	
	IL-2 + IFNa	Sunutinib
Number	18	15
Sex, male	14 (78%)	13 (87%)
Age	68.5 (54–79)	68 (41–76)
Previous nephrectomy	16 (89%)	15 (100%)
Previous partial nephrectomy	2 (11%)	0 (0%)
Previous radiation therapy	1 (6%)	1 (7%)
Hitology, clear cell carcinoma	18 (100%)	13 (87%)
Metastasis site		
Regional lymphnode	0 (0%)	1 (7%)
Distant lymphnode	2 (12%)	2 (13%)
Lung	15 (83%)	11 (73%)
Bone	1 (6%)	4 (27%)
Liver	0 (0%)	0 (0%)
Pancreas	3 (17%)	2 (13%)
Contralateral kidney	1 (6%)	0 (0%)
Others	3 (18%)	0 (0%)
MSKCC risk factors *		
0 (favorable)	5 (28%)	6 (40%)
1–2 (intermediate)	13 (72%)	9 (60%)
>3 (poor)	0 (0%)	0 (0%)

* Risk factors associated with shorter survival according to the Memorial Sloan Kettering Cancer Center (MSKCC) risk class classification are a low serum hemoglobin level, an elevated corrected serum calcium level, an elevated serum lactate dehydrogenase level, a poor performance status, and an interval of less than 1 year between diagnosis and treatment. Abbreviation: Il-2—Interleukin-2; IFNα—Interferon alpha.

**Table 2 cancers-15-02745-t002:** Treatment related adverse events and laboratory abnormalities.

		1st-Line				2nd-Line: Axitinib				
	IL-2 + IFNa (*n* = 18)			Sunitinib (*n* = 15)		(IL-2 + IFNa, *n* = 13)			(Sunitinib, *n* = 5)	
	ALL	>G3	G4	ALL	G3	ALL	>G3	G4	ALL	G3
Total, number (%)	18	8	4	15	13	11	7	1	5	3
	(100)	(44)	(22)	(100)	(87)	(92)	(54)	(8)	(100)	(60)
Adverse event, number										
Chills	3			1						
Decreased appetite	6			3	1	4	1		3	
Diarrhoea				4		2	1		3	1
Dyspepsia				1						
Epistaxis						1				
Face oedema	1									
Fatigue						2	1		3	
Gastric perforation	1	1	1							
Gastritis									1	
Gastrointestinal haemorrhage						1				
Hoarseness						3				
Hypertension	2	1		7	6	9	7		3	2
Hypothyroidism				5		4				
Interstitial lung disease	1									
Lymphopenia				1	1					
Malaise	9			4						
Pain	1									
Palmar-plantar erythrodysaesthesia				8	1	2				
syndrome										
Pancreatitis	1	1	1							
Pleural effusion	1									
Proteinuria	2			2	2	5	1		1	1
Pyrexia	13			5	1					
Stomatitis				4	1				1	
Swelling						1				
Vomiting									1	
Laboratory abnormality number										
Anaemia	4	1		2						
Aspartate aminotransferase increased				1						
AST/ALT ratio abnormal	1									
Blood alkaline phosphatase increased	1	1								
Blood bilirubin increased	1	1								
Blood creatinine increased				2		4	1	1	1	
Hyperglycemia	1	1	1							
Hypoalbuminemia	1									
Hyponatremia				1	1					
Neutrophil count decreased	7	3	1	6	1	1				
Platelet count decreased	3			9	6	1				
White blood cell count decreased	8	3		6	1	1				

Note: Listed are all-treatment related events of interest. All severity was graded according to National Cancer Institute Common Terminology Criteria for Adverse Events, version 4.0. Abbreviation: IL-2—Interleukin-2; IFNα—interferon alpha; AST—aspartate aminotransferase; ALT—alanine aminotransferase. The comparison between the IL-2+IFNα group and the Sunitinib group at 1st line was significant (*p* = 0.027) with the use of Fisher’s exact test applied to the sum of grade 3 and 4 adverse event, however, the grade 4 adverse events (*p* = 0.108) were not significant.

**Table 3 cancers-15-02745-t003:** Best tumor response.

Response	1st-Line			2nd-Line Axitinib		
	IL-2 + IFNa (*n* = 16)	Sunutinib (*n* = 13)	*p*-Value	IL-2 + 1FNa (*n* = 13, 81%)	Sunutinib (*n* = 5, 39%)	*p*-Value
Objective response rate	7 (43%)	4 (31%)	0.474	8 (62%)	0 (0%)	0.019
Complete response	0 (0%)	0 (0%)		1 (8%)	0 (0%)	
Partial response	7 (43%)	4 (31%)		7 (54%)	0 (0%)	
Disease control rate	13 (81%)	15 (100%)	0.099	11 (85%)	3 (60%)	0.261
Stable disease	6 (38%)	9 (69%)		3 (23%)	3 (60%)	
Progressive disease	3 (19%)	0 (0%)		2 (15%)	2 (40%)	

Note: Tumor response was assessed by investigators according to the Response Evaluation Criteria in Solid Tumors. Abbreviation: IL-2—Interleukin-2; IFNα—interferon alpha.

## Data Availability

Research data are stored in an institutional repository and will be shared upon reasonable request to the corresponding author.

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
