# Peer review of "Efficacy and Safety of First-Line Cytokines versus Sunitinib and Second-Line Axitinib for Patients with Metastatic Renal Cell Carcinoma (ESCAPE Study): A Phase III, Randomized, Sequential Open-Label Study"

_cancers, 2023, doi:10.3390/cancers15102745_

Round 1

Reviewer 1 Report

Nowadays Immune-Oncology combination is the standard of care for metastatic renal cell carcinoma patients.
The issue covered by this paper is not interesting as it is outdated. Interleukin shoud not be offered to patients expecially
in clinical trial setting as well
as second line axitinib.

Author Response

We appreciate the reviewer's comments, and we understand the concerns raised. When we initiated this study in 2013, immuno-oncology (IO) drugs were not available, and the standard of care for metastatic renal cancer was being explored. However, with the advent of IO drugs, the role of classical immunotherapy, represented by IL-2+IFNα, has largely diminished. We acknowledge that in randomized controlled trials (RCTs) such as this one, it can take time to obtain results, and by the time conclusions are reached, the standard of care may have changed.

Despite these challenges, we believe it is important to complete the planned study and report the results. The study was designed with the intention of improving long-term outcomes for patients with metastatic renal cancer, and we believe that our findings will still be of value. It is possible that our results could provide insights that are still relevant and meaningful, even in the context of the changing landscape of renal cancer treatment.

In conclusion, we appreciate the reviewer's feedback, and we will carefully consider their suggestions. We are committed to completing our study and reporting our findings in a way that is clear, accurate, and informative.

Reviewer 2 Report

I appreciate colleagues' work, despite the study's negative result. But even negative trials need to be published to help in future research and clinical trial preparations. 

To be precise, it would be OK to mention that updated guidelines for mRCC recommend cabozanitinib as a possible choice for the 2nd line therapy after TKI failure.

From today's perspective, the study's design was ambitious but entirely justifiable at the start of the research and supported with results from previous observations with cytokine therapy for Japan population. Because of the limitation on Japan population, these result should not be translated to the western world. I would recommend adding this limitation, for example, in conclusion. 

Even though the number of patients is minimal, and the sequence evaluation was also affected by the fact that high proportion of patients did not receive axitinib, these data show that the favourable prognostic group is a specific population that needs to be looked at in a completely different way. And we certainly need more research in these area. We also have to wait for long-term follow-ups from TKI-immunotherapy combinations. And in that, I see the contribution of this study, which is another stone in the mosaic of a complex issue.

Although the baseline characteristics are balanced, there is a possible negative factor in the sunitinib group, where a higher proportion of patients had bone mets, which might also have worse outcomes. 

Tables and figures are clear. 

 In my opinion, as the authors admitted to make some conclusions in PFS is tricky. But what I see as an impressive result is the OS where 5year OS is 80 for TKI, which means 10 of 13 patients are still alive. 

In conclusion, I would like to congratulate the authors for their work and effort, which this study had to take, and I recommend the article for publishing.

Author Response

We appreciate the reviewer’s comments. As reviewer’s recommendation, we added the sentences as follows at line 336-337; “Additionally, cabozanitinib is recommended as alternative 2nd-line therapy after TKI failure [20,21].” And, we added the sentences as follows at line 437-438; “In addition, especially in the Japanese population, it may be optional as its effectiveness is better in the Japanese population than in the western population.”

Reviewer 3 Report

Dear Authors, 

thank you for sending this manuscript for evaluation. 

I have a big problem to evaluate this study. At the time when it has been designed it might have produced very interesting results as we knew that immunotherapy had its place in MRCC treatment. Now immunotherapy has regained its place, but in the form of targeted immunotherapy. Has this made classical immunotherapy obsolete? Most oncologists would say yes, but in my opinion IL-2 and IFNalfa can still play a minor role. Its role can rise in the future again as those substances are cheap compared with IO agents and costs of MRCC patient treatment have risen expotetially in last 2 decades. Additionally we have to think about AEs and total OS in patients. Here using IL-2 and IFN alfa can give additional months or years in carefully selected group of patients when followed by TKIs or modern IO agents. 

In this light I think this trial is worth publishing despite its severly  undepowered nature and difficulties with sureness of conclusions. It is a pitty that it has not been finished as designed. In my opinion patients treated in IL-2 and IFNalfa arm would not fare worse than sunitinib patients in terms of total OS. 

Some minor additional remarks:

- please state clearly when the study was designed

- please state hw many institutions were involved

- please explain why immunological treatemant doses and schedules were different 

- please state bioethical approval and signed informed consent

- please explain how you did statistics despite so small numbers of patients. Did you change statistical evaluation plan and methods?

Best ragards

Author Response

We appreciate the reviewer’s comments.

- please state clearly when the study was designed

We added the sentences as follows at line 122-124; “This study was designed and registered at Trial University hospital Medical Information Network (UMIN) Center identifier UMIN 000012522 in July 2013.”

- please state hw many institutions were involved

We mentioned that 10 hospitals in Japan were involved at line 172.

- please explain why immunological treatemant doses and schedules were different

In particular, IL-2 required a hospital visit for intravenous infusion and the administration schedule needed to be adjusted according to the degree of adverse events. To facilitate case registration, the administration schedule was set as a protocol that could be changed at the discretion of the attending physician.

- please state bioethical approval and signed informed consent

We stated Institutional Review Board Statement and Informed Consent Statement at line 447-462.
